# LalaEval: A Holistic Human Evaluation Framework for Domain-Specific Large Language Models

**Chongyan Sun[1,2], Ken Lin[2], Shiwei Wang[2], Hulong Wu[2], Chengfei Fu[2], Zhen Wang[2]**
[1]The Chinese University of Hong Kong [2]Data Science Group, Huolala
sunchongyan@link.cuhk.edu.hk
{adam.lin,three.wang,senter.wu,eden.fu,jackson687.wang}@huolala.cn

## Abstract

This paper introduces LalaEval, a holistic framework designed for the human evaluation of domain-specific large language models (LLMs). LalaEval proposes a comprehensive suite of end-to-end protocols that cover five main components including domain specification, criteria establishment, benchmark dataset creation, construction of evaluation rubrics, and thorough analysis and interpretation of evaluation outcomes. This initiative aims to fill a crucial research gap by providing a systematic methodology for conducting standardized human evaluations within specific domains, a practice that, despite its widespread application, lacks substantial coverage in the literature and human evaluation are often criticized to be less reliable due to subjective factors, so standardized procedures adapted to the nuanced requirements of specific domains or even individual organizations are in great need. Furthermore, the paper demonstrates the framework's application within the logistics industry, presenting domain-specific evaluation benchmarks, datasets, and a comparative analysis of LLMs for the logistics domain use, highlighting the framework's capacity to elucidate performance differences and guide model selection and development for domain-specific LLMs. Through real-world deployment, the paper underscores the framework's effectiveness in advancing the field of domain-specific LLM evaluation, thereby contributing significantly to the ongoing discussion on LLMs' practical utility and performance in domain-specific applications.

## 1 Introduction

The rise of large language models (LLMs) represents a significant step towards artificial general intelligence (Bubeck et al., 2023), showcasing their powerful ability to understand and produce natural language. Although these models have generally been designed for wide-ranging use, one of their most promising applications lies within specific domains such as medicine (Lee et al., 2023; Wang et al., 2023a; Thirunavukarasu et al., 2023), law (Cui et al., 2023) and finance (Li et al., 2023; Wu et al., 2023). For businesses looking to integrate LLMs into their operations, the focus naturally shifts towards models' capabilities in certain industries, as conversations within a particular industry are more prevalent and relevant than broad conversations when in real-world business setting (Guo & Yu, 2022; Zhao et al., 2023).

The process of evaluating these domain-specific models is crucial. Typically, evaluations range from automatic techniques, e.g. Zheng et al. (2024), to human evaluation, with the latter widely regarded as the most comprehensive method. Human evaluation's importance lies in its unparalleled ability to grasp the intricacies of language and context, aspects that automatic methods might miss (Novikova et al., 2017), and more importantly in commercial use, human evaluation also reflects true human preference of businesses' stakeholders, so when developing and comparing LLMs, particularly those meant for domain-specific real-world business use, human evaluation becomes essentially gold standard. It plays a

critical role in measuring how these models perform for commercial use and guides the development process, ensuring the models meet real-world interaction standards.

Given the importance of human evaluation (Chang et al., 2023), creating a standardized framework for human evaluation is vital. This framework, which includes detailed methods for gathering data, setting evaluation rubrics, defining metrics as well as establishing respective evaluation protocols, is essential for ensuring evaluations are consistent, accurate and relevant. It specifically addresses the need for domain-specific LLMs to be closely aligned with actual business requirements, filling the gap between theoretical capabilities and practical application. By focusing on the case study of the logistics domain, our study introduces a detailed evaluation framework designed for a specific domain. This framework is not just an example of the evaluation within this industry, it also showcases a standardized human evaluation framework of domain-specific LLMs in general.

This research makes a meaningful contribution to the discussion on evaluating LLMs, especially for domain-specific uses. By outlining the creation of a human evaluation framework that matches commercial needs, we set a path for a more focused, efficient, and practical use of LLMs in business contexts. Our deployed framework in the logistics domain demonstrates the framework's value to improve the relevance, functionality, and adoption of LLMs in specific domains.

## 2   Related Work

Chang et al. (2023) presents an extensive survey on the evaluation of LLMs, providing a detailed taxonomy of evaluation aspects, including what, where, and how to evaluate. General-purpose models, such as OpenAI's GPT and Google's Gemini, typically do not serve downstream tasks directly. Thus, their evaluation often involves a multifaceted approach (Bang et al., 2023), incorporating a variety of tasks, as demonstrated by Bai et al. (2024); Bian et al. (2023); Liang et al. (2022).

In contrast, the most economically potent applications for commercial, domain-specific use are arguably found in customer operations, i.e. service chatbots (Chui et al., 2023). Accordingly, this paper concentrates on developing an evaluation framework for domain-specific LLMs, particularly for chatbot conversations. Although initially designed for single-round conversations, this framework can be readily extended to multi-round interactions.

There is a significant body of work addressing where to evaluate. Numerous benchmarks and datasets have been established to test the capabilities of LLMs (Hendrycks et al., 2020; Xu et al., 2023; Gu et al., 2023). These efforts provide essential benchmarking datasets and results for mainstream LLMs, facilitating a unified comparison. However, these benchmarks are often not directly applicable to specific company uses due to differences in domains, knowledge, applications, etc. This paper distinguishes itself by offering frameworks and protocols to guide the construction of benchmarks and datasets within commercial organizations for specific domains.

Another stream of literature focuses on how to evaluate. Many benchmarks adopt automatic evaluation methods to obtain results, ranging from traditional metrics (Chin-Yew, 2004) to more LLM-based judgments (Wang et al., 2023b). The advantages of automatic evaluation over human evaluation are clear: it is more standardized and objective, easier to scale, and less costly. However, for non-standard tasks like open generation, human evaluation proves to be more reliable and better aligned with general human preferences (Novikova et al., 2017). Moreover, evaluating domain-specific LLMs often requires profound domain knowledge and even internal organizational knowledge. The human evaluation is typically criticized by its cost and susceptibility to subjective bias, but within the context of commercial use of domain-specific LLMs, a company can often afford the expense of human evaluators and organize effective training. Moreover, domain-specific LLMs typically have large real-world influence (e.g. chatbot serving millions of consumers) and have significant impact on firm operation. In these widespread high-stake circumstances, the resources required are reasonable compared to the potential benefits of ensuring best practice. Nevertheless, there is very little literature offering a holistic framework to systematically guide the implementation

of human evaluation from domain specification and evaluation dataset construction to evaluation rubrics, metrics, and reporting of results. This paper aims to bridge this gap.

# 3   Methodology

## 3.1   Overview

LalaEval encompasses five main components to establish comprehensive protocols guiding the evaluation process of LLMs: (1) *Domain Specification*, (2) *Criteria Establishment*, (3) *Benchmark Dataset Creation*, (4) *Construction of Evaluation Rubrics*, and (5) *Analysis and Interpretation of Evaluation Results*.

1. *Domain Specification*: This component involves defining the scope of specific fields of interest, largely influenced by an organization's goals or objectives with LLMs. It establishes the evaluation process's boundaries.

2. *Criteria Establishment*: This component defines the LLMs' capability dimensions for evaluating performance, effectiveness, or suitability. This ensures that evaluations are based on relevant, objective, and consistently applied measures.

3. *Benchmark Dataset Creation*: This component entails developing standardized tests and compiling carefully curated data collections from scrutinized information sources. It allows for evaluation under consistent conditions, facilitating comparative measurement and analysis.

4. *Construction of Evaluation Rubrics*: This component describes the careful design of grading schemes that detail specific guidelines for measuring various performance aspects. It provides a structured framework to train human evaluators.

5. *Analysis and Interpretation of Evaluation Results*: This component involves systematically examining data collected from the evaluation process to minimize intrapersonal, interpersonal, intramodel, and intermodel variability. It aims to derive meaningful insights and guide decision-making, ensuring the constructive application of outcomes.

## 3.2   Domain specification

In the conceptualization of LalaEval, the first step entails defining the domain that is usually inherently derived from the industries within which an organization operates. The breadth of such an industry definition, however, remains an area for exploration. For instance, should the evaluation scope for a medical LLM include only radiology or extend to the broader field of medicine? Should the focus for a financial LLM be confined to bonds, or should it cover the wider realm of capital markets?

We propose a hierarchical structuring of subdomains for LalaEval, which organizes these subdomains by their relative significance, ranging from narrow to broad scopes. Employing backward induction, we begin with a highly specific subdomain pertinent to the organization, progressing towards a more generalized industry definition. By adhering to the principle of mutual exclusivity and collective exhaustiveness, we enumerate the most granular subdomains and their parallel subdomains, progressively ascending to encompass broader subdomains. This process is iterated multiple times until a sufficiently broad domain scope is achieved.

Subsequently, we employ qualitative prioritization to define both the scope and the hierarchical significance of each subdomain. This prioritization may adopt either a linear progression from the most specific to the most broad domain or a tree-like structure that includes parallel subdomains at various levels, contingent upon business imperatives.

Figure 1 presents our case by hierarchical domain specification in the logistics industry, with the final chosen subdomains highlighted. We establish both the scope and priorities for subdomains, ranking them from high (P0) to low (P3) priority: P0 (Intracity Freight

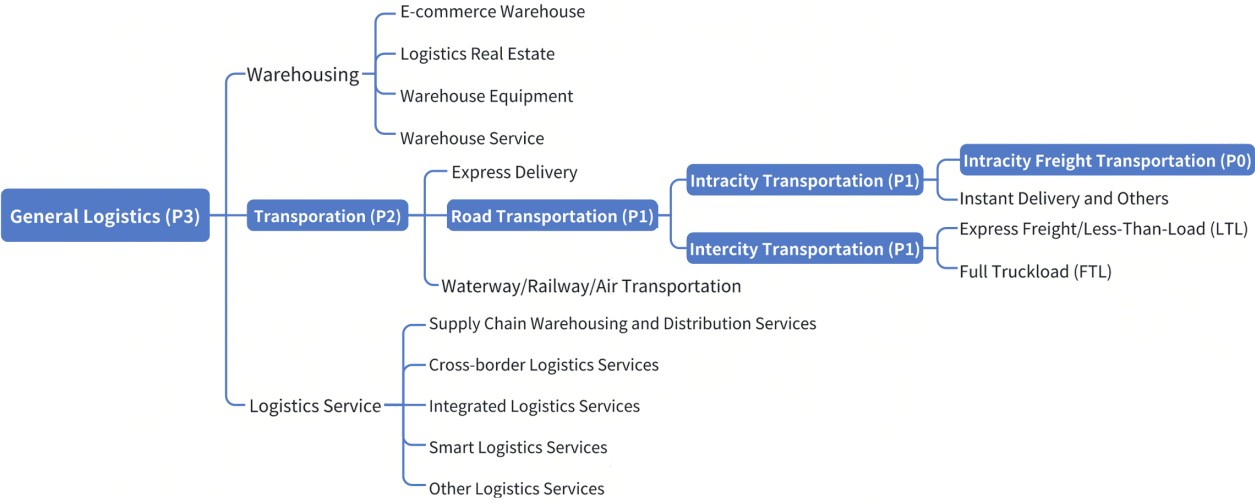

Figure 1: The hierarchical domain specification in the logistics industry

Transportation), P1 (Road Transportation; Intracity and Intercity Transportation), P2 (Transportation), and P3 (Logistics). This prioritization informs the overall evaluation process, emphasizing the critical importance of intracity freight transportation in our case while also acknowledging the integral roles of road transportation and general logistics.

This hierarchical specification of subdomains and their prioritization not only informs but also directs the subsequent components of LalaEval.

## 3.3 Criteria Establishment

### 3.3.1 General Capability

The objective of establishing criteria for general capabilities is to evaluate the performance of LLMs a wide array of natural language tasks not confined to any specific domain. This includes their capability in comprehending and generating natural language, recognizing contextual cues, sustaining coherence throughout conversations, and processing and conveying information with accuracy.

The rationale for evaluating such general capabilities includes (1) *Foundation for domain capability*: A solid foundation in general capability is crucial for the performance of any LLM, providing the groundwork upon which domain capability is constructed. A lack in general language capability can impede LLMs' capability to understand or generate responses accurately within specialized domains. (2) *Flexibility and Adaptability*: LLMs that exhibit strong general language capabilities demonstrate enhanced flexibility and adaptability. This is vital for applications requiring comprehension of inputs from diverse domains or the integration of new knowledge without the need for extensive retraining. (3) *Understanding and Reasoning*: Evaluating general capabilities aids in identifying LLMs' capacity for understanding complex queries, reasoning, and generating coherent responses that are contextually appropriate. These attributes are indispensable for practical applications.

To systematically evaluate these general capabilities, we have adapted from (Xu et al., 2023) and outlined dimensions of general capabilities in the following, and we also show example questions and defined difficulty levels in Appendix A.

- *Semantic Understanding*: Crucial for LLMs is understanding the basic meaning of language, and special terms such as idioms and cultural references, to ensure meaningful user interaction. Evaluation involves questions that test understanding across linguistic occasions.

- *Contextual Conversation*: LLMs need to remember past interactions to maintain coherent conversations, essential for sustained engagement in customer service or conversational applications. Multi-round questions can evaluate this skill.

- *Answer Completeness and Coherence*: Responses must be clear, concise, and address queries directly, ensuring outputs are practical and user-centric, especially in informative or decision-support settings.

- *Factuality*: Accuracy in responses, especially for questions expecting definitive answers, is critical. This is vital in industries like logistics where errors can have significant consequences. Verification against trusted sources evaluates this aspect.

- *Creativity*: The ability to generate creative content, such as marketing material or innovative responses, highlights LLMs' capacity for engaging content creation. This is evaluated through creativity requests with certain constraints.

- *Logical Reasoning*: Tasks requiring numerical or logical deduction evaluate this capability, relevant for problem-solving in industries like logistics. Evaluation includes mathematical or logic puzzles.

### 3.3.2 Domain Capability

The domain capability focuses on LLMs' expertise within certain industries, evaluating its understanding of domain-specific terminologies, concepts, regulations, and operational nuances. It also evaluates LLMs' capability to provide insightful, accurate responses to domain-specific queries.

The rationale for evaluating domain-specific capability is (1) *Domain-Specific Performance*: The principal motive is to ascertain the efficacy of LLMs in logistics-specific applications. It is imperative for LLMs to not merely comprehend general language but also to exhibit a profound understanding of specialized domain knowledge and intricacies. (2) *Practical Usability*: LLMs endowed with strong logistics domain capabilities hold greater practical value for industry professionals and businesses. Such LLMs are capable of providing more accurate and relevant insights, thereby facilitating enhanced decision-making processes and operational efficiency. (3) *Customized Solutions*: Evaluating this capability allows for the development of more customized, domain-specific solutions that can address specific challenges and needs within the logistics domain, providing a competitive edge to businesses leveraging these LLMs.

Focusing on the dimensions of factuality and creativity, we have outlined and illustrated subdimensions of capabilities in logistics domain in the following, and we also show example questions and defined difficulty levels in Appendix B.

- *Conceptual and Terminological Understanding*: Knowledge of specific terms and concepts is fundamental in accurately interpreting and responding to industry-related queries. This necessitates an evaluation that includes questions derived from the logistics domain's lexicon and operational nuances.

- *Company Information*: LLMs should be conversant with key players in the industry and relevant corporate data, reflecting the model's utility as an informative resource. Questions about major companies, their operations, and market positions can evaluate this aspect.

- *Legal and Policy Knowledge*: Given the regulatory environment surrounding logistics operations, LLMs must be adept at navigating legal and policy-related queries. This can be evaluated through questions that require the LLM to reference specific regulations or guidelines applicable to logistics.

- *Industry Insights*: The ability to provide informed opinions or data about the logistics market, trends, and future outlooks showcases LLMs' depth of knowledge and analytical capabilities. Crafting scenarios that ask for analysis or predictions based on current data evaluates this competency.

- *Company-specific Knowledge*: For LLMs deployed by specific companies, such as Huolala, understanding the company's services, history, and strategic vision is

crucial. This ensures the LLM can serve as an effective chatbot. Questions tailored to the company's operations and strategy evaluate this knowledge.

- *Creative Capability in Logistics Context*: Beyond generic creativity, the ability to generate content specifically tailored to the logistics industry's stylistic and contextual requirements is valuable for marketing and customer engagement purposes. Questions requiring domain-specific creative responses can evaluate this capability.

## 3.4 Benchmark Dataset Creation

The primary goal of establishing a high-quality benchmark dataset is to develop an evolving bank of question-answer (QA) pairs for human evaluation. Existing public benchmarks and datasets are not utilized due to (1) the absence of benchmarks tailored to the specific domain; (2) the proprietary benchmark's closer alignment with company-specific needs; and (3) the lack of continuous updates in most public datasets, which fails to meet the timely knowledge requirements for the business application of LLMs. Concurrently, this dataset aims to include a broad spectrum of capabilities outlined in Section 3.3, thereby ensuring a uniform evaluation of LLMs' performance across diverse metrics.

The benchmark dataset are developed through the process outlined in following steps:

1. *Accumulation of the Raw Corpus*: Guided by the previous components, we systematically collect and compile original texts and corpora from verified sources. This repository of raw corpus forms the groundwork for generating insightful and pertinent QA pairs.

2. *Production of QA Pairs*: (1) Development of the Question Plan: Formulate a structured plan outlining the desired number of QA pairs, categorized by various levels of difficulty and capabilities. This plan should be iterative to allow for the continuous enhancement of the dataset. (2) Selection of Question Designers: Identify and appoint individuals responsible for question design, ensuring they possess an in-depth understanding of the evaluation framework and access to the raw corpus. (3) Creation of QA Pairs: The question designers will distill relevant information from the raw corpus or other authoritative sources to craft QA pairs. Each pair will include a question and its corresponding standard answer, for instance, "Q: How many hours are there in a day? A: 24 hours." Importantly, the source of information for each QA pair must be documented to maintain traceability and credibility.

3. *Quality Inspection and Database Entry*: Following the generation of QA pairs, a thorough quality inspection is conducted to verify that the pairs adhere to predefined criteria. QA pairs that pass this inspection are incorporated into the dataset, whereas those failing to meet the standards are redirected back to the designers for refinement.

The process of proposing QA pairs for the benchmark dataset is a critical component in the evaluation of LLMs, particularly within specific domains such as the logistics industry. Through a systematic approach to compiling raw corpora, creating relevant QA pairs, and executing rigorous quality inspections, this initiative seeks to establish a comprehensive dataset. This dataset not only lays the foundation for a thorough evaluation of LLMs but also facilitates the continuous improvement and benchmarking of LLMs, thereby ensuring their relevance and applicability in real-world scenarios.

## 3.5 Construction of Evaluation Rubrics

The objective of the construction of evaluation rubrics is to systematically evaluate the performance of various LLMs using the benchmark dataset from Section 3.4. This process is designed to guide the training of human evaluators and ensure more consistent outcomes, both intraperson and interperson.

The general evaluation scale is 0-3 points. If a response contains any incorrect information, it will get 0 points. 1-3 points measure the degree of correctness, completeness, creativity etc. Meanwhile, special consideration is given to the timeliness of responses, recognizing the

importance of current knowledge and the LLMs' ability to reflect the most recent information or anticipate future developments. Appendix C presents the preceding general grading principle and the special consideration about timeliness.

The evaluation rubrics for evaluating the general capabilities and logistics domain capabilities of LLMs are comprehensively outlined in Appendices D and E, respectively.

### 3.6 Analysis and Interpretation of Evaluation Outcomes

After trained human evaluators are ready (see Appendix G for details of human evaluator training process), we randomly draw a subset from the dataset based on agreed difficulty level and quantities in the grading process. To keep integrity, we employ a single-blind procedure, wherein model responses to the QA pairs are anonymized and presented in a randomized order to a panel of at least three human evaluators. This procedure mitigates bias, ensuring that human evaluators cannot infer the origin of the responses, thus maintaining objectivity. Table 1 shows the demo of human evaluators' interface.

|  | Standard Answer | Grading Principle | Position 1 | Position 2 | Position 3 | Position 4 |
|---|---|---|---|---|---|---|
| **Question 1** | [Question content] | [Principle content] | Model 4 Response | Model 1 Response | Model 2 Response | Model 3 Response |
| **Question 2** | [Question content] | [Principle content] | Model 3 Response | Model 1 Response | Model 4 Response | Model 2 Response |

Table 1: The demo of human evaluators' interface

The evaluation results are systematically compiled into a four-dimensional table that encapsulates the evaluated capabilities, question numbers, human evaluator IDs, and grades allocated to each LLM for every question like Table 2. This structured data allows for analysis across different dimensions of model performance. Grades for each model within specific capabilities are aggregated and normalized to 100 points, offering a detailed view of model capabilities. Moreover, more comprehensive grading can be achieved through the weighted aggregation of detailed grades, reflecting the LLMs' overall performance spectrum.

| Capability Dimension | Question Number | Evaluator Number | Model 1 Grade | ... | Model $q$ Grade | ... | Model $Q$ Grade |
|---|---|---|---|---|---|---|---|
| Dimension 1 | Question 1 | Evaluator 1 | $AS_{111}$ | $\cdots$ | $AS_{q11}$ | $\cdots$ | $AS_{Q11}$ |
| Dimension 1 | Question 1 | Evaluator 2 | $AS_{112}$ | $\cdots$ | $AS_{q12}$ | $\cdots$ | $AS_{Q12}$ |
| $\vdots$ | $\vdots$ | $\vdots$ | $\vdots$ | $\ddots$ | $\vdots$ | $\ddots$ | $\vdots$ |
| Dimension $j$ | Question $k$ | Evaluator $i$ | $AS_{1ki}$ | $\cdots$ | $AS_{qki}$ | $\cdots$ | $AS_{Qki}$ |
| Dimension $j$ | Question $k$ | Evaluator $i+1$ | $AS_{1k(i+1)}$ | $\cdots$ | $AS_{qk(i+1)}$ | $\cdots$ | $AS_{Qk(i+1)}$ |
| $\vdots$ | $\vdots$ | $\vdots$ | $\vdots$ | $\ddots$ | $\vdots$ | $\ddots$ | $\vdots$ |
| Dimension $J$ | Question $K_J$ | Evaluator $n$ | $AS_{1(K_J)n}$ | $\cdots$ | $AS_{q(K_J)n}$ | $\cdots$ | $AS_{Q(K_J)n}$ |

Table 2: Four-dimensional table of evaluators' grading results. $AS_{qki}$ represents the grade given by evaluator $i$ to the response of the $q_{th}$ model for the $k_{th}$ question.

The grade calculation for model $q$ involves single dimension $j$ grade calculation using the formula $Grade(qj) = \frac{\sum_{k=1}^{K_j} \sum_{i=1}^{n} AS_{qki}}{\sum_{k=1}^{K_j} \sum_{i=1}^{n} TS_k}$ , then aggregating across all dimensions to compute the total grade as $Grade(q) = \sum_{j=1}^{J} w_j Grade(qj)$ . This process evaluates a LLM's capability across various dimensions and overall, by weighing each dimension's grade, equally or otherwise. Reporting includes both the grades for different dimensions and the total grade, facilitating a comprehensive assessment of each LLM's performance. Repeating these steps for all models under evaluation provides a comparative analysis of their strengths and weaknesses.

Our methodology also aims to reduce the subjective factors of human evaluators as much as possible, incorporating automated dispute analysis and checks for grade stability and reliability. This phase is crucial for identifying low-quality grades and questions and

attributing grading fluctuations to transparent factors, such as evaluator inconsistency or changes in LLM responses. The detailed dispute analysis and grade fluctuation analysis methods are described in Appendix F.

## 3.7 Overall Deployment Structure

After describing the five main conceptual components of LalaEval, we now turn to the practicalities of deployment, presenting an approach that brings our framework into operational reality, ensuring a comprehensive evaluation process for domain-specific LLMs. This integration strategy, designed to foster standardization, as well as efficiency and adaptability, is succinctly illustrated in Figure 2. Through this streamlined depiction, the operational blueprint that guides the application of LalaEval is presented, facilitating a clear understanding of its modular architecture and the dynamic interactions essential for evaluation. The results presented in Section 4 are also generated using this deployment strategy.

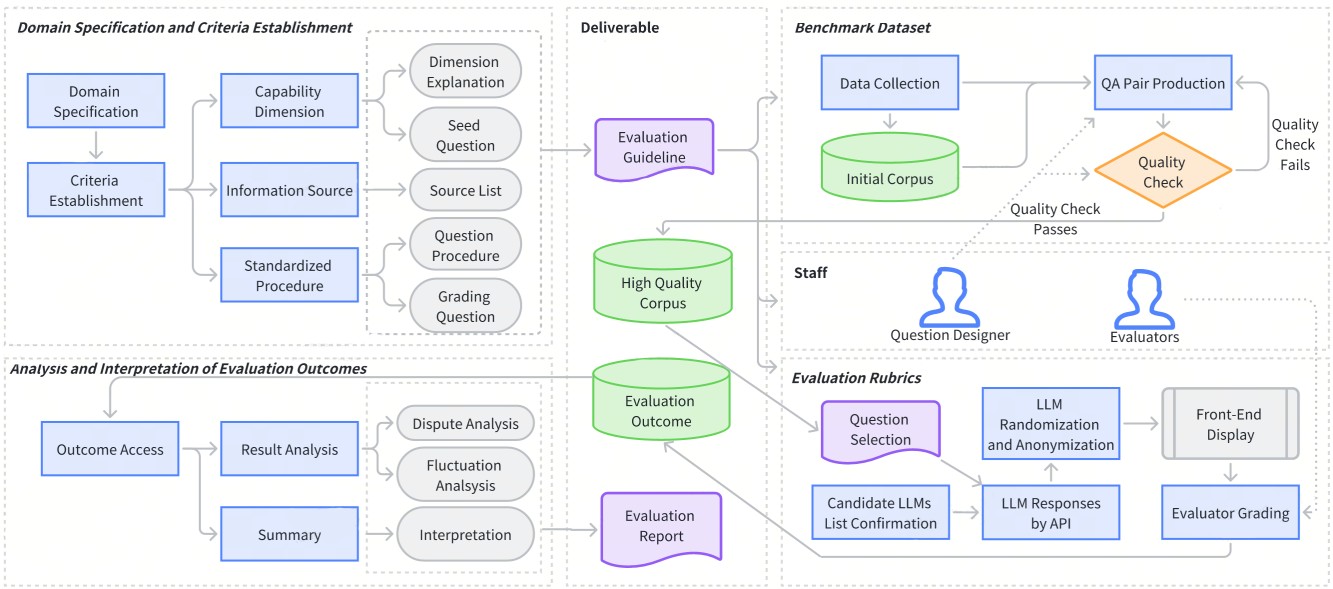

Figure 2: The overall deployment diagram of LalaEval

## 4 Results

To demonstrate the effectiveness and the practical application of LalaEval, we present the zero-shot evaluation results of various LLMs including OpenAI's GPT-4 (without web access), Baidu's Ernie Bot (with web access), and our proprietary LLM v1, v2, and v3 (PLLM1/2/3). PLLM1/2/3 are iterations fine-tuned from the ChatGLM2-6B foundation, incorporating web access, retrieval-augmented generation, or a combination of both, respectively.

Our evaluation is demonstrated in two distinct result sets: (1) The accuracy of the LLMs (Table 3), where we present the percentage of responses achieving non-zero grades across different capability dimensions. This metric serves as a direct indicator of each LLM's capability to generate relevant and accurate response. (2) The normalized average grades (Table 4) for each LLM across capability dimensions, providing more quantitative view of LLM performance. The complete results of grades and disagreement by all capability dimensions are included in Appendix H. The results reflect various levels of coverage of capability dimension defined in Section 3.3 where *Domain-Factuality* is defined as the average

capability within the domain except creative capability, enabling finer-grained comparison across LLMs. The results only represent evaluation under our LalaEval framework, focusing more on capabilities in the logistics domain.

| Capability Dimension | GPT-4 | Ernie Bot | PLLM3 | PLLM2 | PLLM1 |
|---|---|---|---|---|---|
| Domain-Factuality | 45.0% | 84.0% | 93.2% | 92.0% | 86.2% |
| Domain | 54.2% | 86.7% | 87.0% | 86.0% | 81.2% |
| General | 80.0% | 89.3% | 68.0% | 63.0% | 62.7% |
| Overall | 67.1% | 88.0% | 77.5% | 74.5% | 71.9% |

Table 3: The accuracy of evaluated LLMs by narrower to broader capability dimension coverage

| Capability Dimension | GPT-4 | Ernie Bot | PLLM3 | PLLM2 | PLLM1 |
|---|---|---|---|---|---|
| Domain-Factuality | 38.8 | 79.7 | 88.7 | 81.4 | 90.3 |
| Domain | 48.1 | 81.8 | 80.6 | 74.5 | 81.9 |
| General | 77.0 | 87.1 | 59.4 | 59.1 | 63.6 |
| Overall | 62.6 | 84.4 | 70.0 | 66.8 | 72.7 |

Table 4: The normalized average grades of evaluated LLMs by narrower to broader capability dimension coverage

The comparative analysis, as illustrated in Tables 3 and 4, highlights a landscape of LLM performance across different capabilities. GPT-4 and Ernie Bot exhibit closely matched strengths in general capabilities, significantly outperforming the proprietary LLMs. However, the narrative shifts within the logistics domain, where proprietary LLMs, alongside Ernie Bot, demonstrate superior performance over GPT-4, possibly due to the lack of information used to fine-tune proprietary LLMs. Notably, within this domain, the proprietary LLMs marginally surpass Ernie Bot in factuality, showing their refined capability. These insights serve not only to benchmark the current state of LLMs within the company but also to direct future development efforts.

## 5 Future Work

LalaEval may have some potential limitations, including evaluator subjectivity, data selection bias, dynamic changes in domain knowledge, and scalability concerns. To address these challenges and further enhance LalaEval's effectiveness, several avenues for future work can be proposed.

- *Standardized Training and Assessment:* Beyond procedures described in Appendix G, exploring more structured training protocols across domains can further improve the reliability and generalizability of LalaEval.
- *Enhanced Dataset Representation:* Ensuring the benchmark dataset is comprehensive and representative in the domain can minimize biases arising from data selection. This step involves continuous refinement and expansion of dataset sources to cover diverse scenarios.
- *Adaptability to Dynamic Domains:* Given the evolving nature of domain-specific knowledge, continuous updates to evaluation protocols and benchmarks are essential. This adaptive approach helps LalaEval stay relevant and effective in terms of dynamic industry changes.
- *Automation Integration:* Supplementing human evaluation with automated methods and support systems can streamline the evaluation process. Automation can assist in handling large volumes of data and tasks, enhancing scalability while maintaining evaluation reliability.

- *Robust Support Systems:* Developing robust support systems to assist evaluators in interpreting ambiguous cases or complex scenarios can further improve evaluation consistency and reliability.

## 6 Conclusion

This study introduces LalaEval, a novel framework for evaluating domain-specific LLMs, with a focus on standardizing human evaluations. By detailing a comprehensive methodology spanning domain specification to results analysis, LalaEval addresses the critical gap in the standardized human evaluations of domain-specific LLMs, exemplified through its application in the logistics domain.

LalaEval's deployment showcases its capability to illuminate performance differences among LLMs, guiding model selection and development for domain-specific applications. LalaEval not only advances the field of LLM evaluation by establishing end-to-end human evaluation protocols but also emphasizes the importance of aligning LLM capabilities with practical needs.

LalaEval is also a general human evaluation framework which are disentangled with any specific domain. The results of only logistics domain are reported because LalaEval was developed and finalized when exploring standardized human evaluation framework in this domain. LalaEval has been applied to other domains such as HR/IT maintanance/text-to-SQL/telemarketing and brought business value. It is believed those who are interested in employing LalaEval to evaluate other domain-specific LLMs can easily implement it.

In summary, LalaEval represents a significant step forward in the evaluation of domain-specific LLMs, offering a structured and human-centric approach to understanding LLM performance. Its contributions lay the groundwork for future research and application of LLMs across various domains, highlighting the evolving need for evaluation methodologies that are as dynamic and specialized as the LLMs they seek to evaluate.

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

# A    Details of General Capability

| Capability Dimension | Definition | Difficulty Level Definition | Example Question |
|---|---|---|---|
| **Semantic Understanding** | The capability to correctly understand the content of the question and possible proverbs and idioms, etc. | *Simple:* Questions are straightforward and easy to understand.
*Intermediate:* Questions have a higher level of difficulty, such as involving metaphorical meanings.
*Difficult:* Questions involve Chinese cultural context. | *Simple:* Introduce Shenzhen.
*Intermediate:* What are the two clouds in physics?
*Difficult:* What does "Luoyang Paper Expensive" mean? |
| **Contextual Conversation** | The capability of memory and contextual connection in multi-round conversations | *Simple:* Referring to the previous round of answers or questions.
*Intermediate:* Referring to answers or questions from the last 3 rounds.
*Difficult:* Repeating questions or deviating from the topic before returning to it. | NA |
| **Response Completeness and Coherence** | Consistency in language, no garbled characters, no interruption, no repeated answers and complete response | *Simple:* Input in Chinese.
*Intermediate:* Mixed input with English and Traditional Chinese.
*Difficult:* Inputs containing multiple questions. | *Simple:* Introduce yourself
*Intermediate:* The main character is called Mike. Please write a short story.
*Difficult:* [Three different questions] |
| **Factuality** | Contradiction to objective facts or hallucination | *Simple:* Can be obtained through information retrieval, mainly common sense questions.
*Intermediate:* Requires multiple retrievals or a large amount of information.
*Difficult:* Requires analyzing a lot of information and making judgments. | *Simple:* Which side does the sun rise from?
*Intermediate:* List the emperors of the Ming Dynasty and briefly introduce their main achievements and failures.
*Difficult:* Which company had better revenue growth last year, Microsoft or Google? |
| **Creativity** | Generate text that meets the required style, such as quatrains, poems and slogans | *Simple:* High degree of freedom in creation.
*Intermediate:* Creation with 1–2 constraints.
*Difficult:* Creation with at least 3 constraints. | *Simple:* Write a poem.
*Intermediate:* Write a poem about spring.
*Difficult:* Write a seven-character poem about spring. |
| **Logical Reasoning** | Mathematical problems and logical relations | *Simple:* Elementary simple arithmetic.
*Intermediate:* Involve junior high school level mathematics.
*Difficult:* Involve high school mathematics or identify wrong/unanswerable questions. | *Simple:* $1 + x = 3$, what is x?
*Intermediate:* The chicken and rabbit in the same cage problem.
*Difficult:* Permutations and combinations. |

Table 5: Definition, difficulty level and examples questions of general capability

# B Details of Logistics Domain Capability

| Capability Dimension | Subdimension | Definition | Difficulty Level Definition | Example Question |
|---|---|---|---|---|
| | Conceptual and Terminological Understanding | Define the basic concepts and terminology of the logistics domain. Focus on various sub-domains related to the logistics industry, and answer more specialized questions. | *Simple:* Accessible through common queries, primarily involving basic conceptual questions in the logistics domain. *Intermediate:* Requires multiple information retrievals, or the use of more specialized (beyond simple internet searches) information retrieval channels, or involves a large amount of information. *Difficult:* Necessitates numerous queries and a large volume of information, or involves special querying methods, or requires processing such as comparative summarization and analysis of results. | *Simple:* What is a refrigerated truck? / What is intracity freight transportation? / What are the main types of vehicles used in intracity freight transportation? *Intermediate:* What are the fee standards for different types of vehicles in intracity freight transportation? *Difficult:* What are the similarities and differences between tracking technologies in intracity freight transportationand the courier industry? |
| | Company Information | Answer questions related to the major companies in the logistics industry (or its upstream and downstream sectors). | *Simple:* Accessible through common queries, primarily involving basic facts about the company with minimal information required. *Intermediate:* Requires multiple searches, or the use of more specialized (beyond simple internet searches) information retrieval channels, or involves a significant amount of information. *Difficult:* Necessitates numerous searches and a substantial amount of information, or involves unique querying methods, or requires processing such as comparative summarization and analysis of results. | *Simple:* The IPO date of Company A. *Intermediate:* The revenue composition of Company B in 2023. *Difficult:* A comparison of the revenue growth rates of Company A and B in 2023. |
| | Legal and Policy Knowledge | Answer questions related to market regulation and legal regulations in the logistics industry. | *Simple:* Accessible through common queries, involving introductions to laws and regulations, such as the main application areas of these laws and regulations. *Intermediate:* Requires multiple information searches and combinations, or involves specific legal provisions. *Difficult:* The application of specific laws and regulations in real-world scenarios. | *Simple:* What are freight-related laws in China? *Intermediate:* What are the truck restriction rules in Shenzhen? *Difficult:* Given [a specific real-world scenario], can you provide the corresponding legal provisions? |
| | Industry Insights | Answer questions about the macro structure, current status, and development of the logistics industry. | *Simple:* Accessible through common queries, mainly concerning basic industry knowledge and information. *Intermediate:* Requires multiple information retrievals, or the use of more specialized (not just simple internet searches) information retrieval channels, or entails a significant amount of information, or involves some level of analysis. *Difficult:* Based on available information, it necessitates combined analysis, forecasting, and similar methods to acquire. | *Simple:* Which are the leading companies in the courier industry? *Intermediate:* What is the market size distribution of the leading companies in the courier industry? *Difficult:* What are the future development prospects of Company A in the courier industry? |
| | | Answer questions related to Huolala. | *Simple:* Accessible through common queries, mainly involving overall knowledge of the company without delving into specific business details. *Intermediate:* Requires multiple information searches, or the use of more specialized (beyond simple internet searches) information retrieval channels, or involves a large amount of information. *Difficult:* Information retrieval involves more specialized methods, and requires combined analytical reasoning. | *Simple:* When was Huolala founded? *Intermediate:* What was Huolala's revenue in 2022? *Difficult:* What are the future development prospects of Huolala's service line A? |
| | | From the user's perspective: answer questions related to Huolala's services posed by users. | *Simple:* Accessible through common queries, primarily involving basic business questions. *Intermediate:* Involves multiple information retrievals, mainly for a more detailed breakdown and understanding of business questions. *Difficult:* Requires a certain level of inference and judgment in combination with business knowledge. | *Simple:* What types of vehicles are available for moving home in Shenzhen? *Intermediate:* What types of costs are associated with moving home in Shenzhen? *Difficult:* What type of vehicle is suitable for a typical 3-person family to use for moving home? |
| Factuality | Specific Knowledge about Huolala | From the driver's perspective: answer questions related to Huolala's services posed by drivers. | *Simple:* Accessible through common queries, primarily involving basic business questions. *Intermediate:* Involves multiple information retrievals, mainly for a more detailed breakdown and understanding of business questions. *Difficult:* Requires a certain level of inference and judgment in combination with business knowledge. | *Simple:* What are the steps for a driver to accept a job? *Intermediate:* What are some important considerations for drivers during the job acceptance process? *Difficult:* How can a driver receive more jobs? |
| Creativity | Creative Capability in Logistics Context | Generate scripts for marketing, advertising, promotions, and market research | *Simple:* Basic notification scripts. *Intermediate:* Marketing, advertising, and promotional copy that involves more creativity. *Difficult:* Scripts for user research, customer service, and other complex communication scenarios. | *Simple:* We need to send a text message to users about a coupon. Please help design a script. *Intermediate:* September is the company's home-moving discount season, please help design promotional script for potential home-moving customers. *Difficult:* We need to conduct surveys through phone calls with drivers who have stopped using the platform, please help design a script to make the drivers more willing to participate. |

Table 6: Definition, difficulty level and examples questions of logistics domain capability

## C Grading Principle

C.1 General Grading Principle

Factual Questions:

- If the response contains incorrect information, grade 0.

- If the response is correct but less complete than the standard answer, grade 1.

- If the response is fully consistent with the standard answer's points, grade 2.

- If the response is fully consistent with the standard answer's points and provides additional correct information, grade 3.

Open-ended Questions:

- If the answer contains incorrect information, grade 0.

- In the absence of errors, the answer is judged by evaluators based on depth and breadth compared to the standard answer to grade 1 or 2.

C.2 Special Consideration

Timeliness of Responses:

- If the response does not specify a clear time but matches the standard answer, it is considered correct.

- If the response does not specify a clear time and does not match the standard answer, it is considered a vague answer and judged incorrect.

- If the response includes explicit time that is different from the question's time, but the response would be correct from the given time point, it is considered correct.

## D Grading Rubrics of General Capability

| Capability Dimension | Rubrics |
| --- | --- |
| Semantic Understanding | 0: Incorrect understanding of the question
1: Correct Understanding of the question |
| Contextual Conversation | 0: No contextual conversation capability
1: Limited contextual conversation capability
2: Good contextual conversation capability
3: Excellent contextual conversation capability |
| Response Completeness and Coherence | 0: Incomplete response or inconsistent language
1: Complete |
| Factuality | 0: Incorrect information
1: Correct information but incomplete
2: Correct information and complete |
| Creativity | 0: No consistency with requirements
1: Limited consistency with requirements
2: Complete consistency with requirements and limited artistic conception
3: Complete consistency with requirements and stronger artistic conception |
| Logical Reasoning | 0: Incorrect result
1: Correct result |

Table 7: Grading rubrics of general capability evaluation

# E   Grading Rubrics of Logistics Domain Capability

| Capability Dimension | Subdimension | Rubrics |
|---|---|---|
| | Conceptual and Terminological Understanding | 0: Incorrect information
1: Correct information but incomplete
2: Correct information and complete |
| | Company Information | 0: Incorrect information
1: Correct information but incomplete
2: Correct information and complete |
| Factuality | Legal and Policy Knowledge | 0: Incorrect information
1: Correct information but incomplete
2: Correct information and complete |
| | Industry Insights | 0: Incorrect information
1: Correct information but incomplete
2: Correct information and complete |
| | Company-specific Knowledge | 0: Incorrect information
1: Correct information but incomplete
2: Correct information and complete |
| Creativity | Creative Capability in Logistics Context | 1: Limited consistency with requirements
2: Complete consistency with requirements but not in logistics context
3: Complete consistency with requirements and in logistics context |

Table 8: Grading rubrics of logistics domain capability evaluation

# F   Dispute Analysis and Grade Fluctuation Analysis

## F.1   Dispute Analysis

1. Evaluator Dispute: automatically identifies potential low-quality grades and evaluators.

   (a) Define the grading dispute decision function $F_{ikq}$: This function takes the value 0 or 1. When the $i_{th}$ grader has a dispute over the grade for the $q_{th}$ model on the $k_{th}$ question (QA pair $kq$), $F_{ikq} = 1$, otherwise $F_{ikq} = 0$. The criterion for determining a dispute can be a manual rule. For example: if evaluator $i$'s grade $> 0 (= 0)$, but all other graders' grades $= 0 (> 0)$, it is defined that grader $i$ has a dispute over the QA pair, assigning $F_{ikq}$ as 1, otherwise, it is considered as no dispute, assigned as 0.

   (b) Calculate the dispute level of grade $i$ on dimension $j$: $C(ij) = \frac{\sum_{k=1}^{K_j} \sum_{q=1}^{Q} F_{ikq}}{K_j Q}$. $Q$ represents the number of LLMs evaluated. This signifies the proportion of QA pairs that grade $i$ has disputed grades within that dimension.

   (c) Calculate the overall dispute level of grader $i$: $C(i) = \sum_{j=1}^{J} w_j C(ij)$. Here $J$ represents the number of dimensions, $w_j$ the weight of dimension $j$, and $\sum_{j=1}^{J} w_j = 1$. If no particular dimension is of special interest, all dimensions can be equally weighted.

   (d) Repeat the previous steps to calculate the dispute level for all graders.

   (e) List the total dispute level of all graders, dispute levels in different dimensions, and identify graders with high dispute levels. Review the disputed grades of high-dispute graders, and if a manual review confirms significant issues with their grades, conduct retraining for the disputed graders and consider invalidating their grading results.

2. Question Dispute: automatically identifies potential low-quality QA pairs.

   (a) Define the question dispute decision function $G_{kq}$: This function takes the value 0 or 1. When there is a divergence among graders' grades for the $q_{th}$ model on the $k_{th}$ question (QA pair $kq$), $G_{kq} = 1$, otherwise $G_{kq} = 0$. The criterion for determining dispute can be a manual rule. For example: when half of the graders' grades $> 0 (= 0)$, and the other graders' grades $= 0 (> 0)$, the QA pair is considered to have divergence, assigning $G_{kq}$ as 1, otherwise it is considered non-dispute, assigned as 0.

(b) Calculate the dispute level of question $k$: $C(k) = w_1 \times (\sum_{q=1}^{Q} G_{kq}) + w_2 \times \frac{\sum_{i=1}^{n} \sum_{q=1}^{Q} F_{ikq}}{n}$, where $w_1 + w_2 = 1$, $F_{ikq}$ is the grading dispute decision function. The reason for including grading dispute is that it represents a certain level of fluctuation. If grading disputes are not to be considered, $w_2$ can be set to a lower weight, or simply let $w_2 = 0$.

(c) Calculate the dispute level for all questions: Repeat steps 1-2 for each question.

(d) List the top $N$ disputed questions, manually determine if there are issues with the questions or standard answers. If issues are indeed present, invalidate the question and conduct retraining for the question designers on the standard of question design.

## F.2 Grade Fluctuation Analysis

If the same LLM shows significant grade changes between two rounds of evaluation, the changes must be attributed and explained.

1. Breakdown of Fluctuation Causes: grading fluctuations can be broken down into 4 reasons (1) changes in the question itself (2) changes in the LLM's response (3) inconsistency in the same evaluator's grading (4) changes in evaluators.

2. Tagging Questions and LLM Responses: Manual tagging is required to determine if a question has changed; if the question remains the same, then determine if the LLM's response has changed (if the information in the response is consistent, it is considered the same, exact sameness is not required)

3. Single Dimension Grade Breakdown: Based on whether the Q and A of the QA pair are the same and whether the evaluator is the same, the fluctuation can be broken down into 6 scenarios shown in Figure 3. Calculating the differences in grades before and after for these 6 scenarios and aggregating them can quantify the contributions of the 4 fluctuation reasons to the overall grade change.

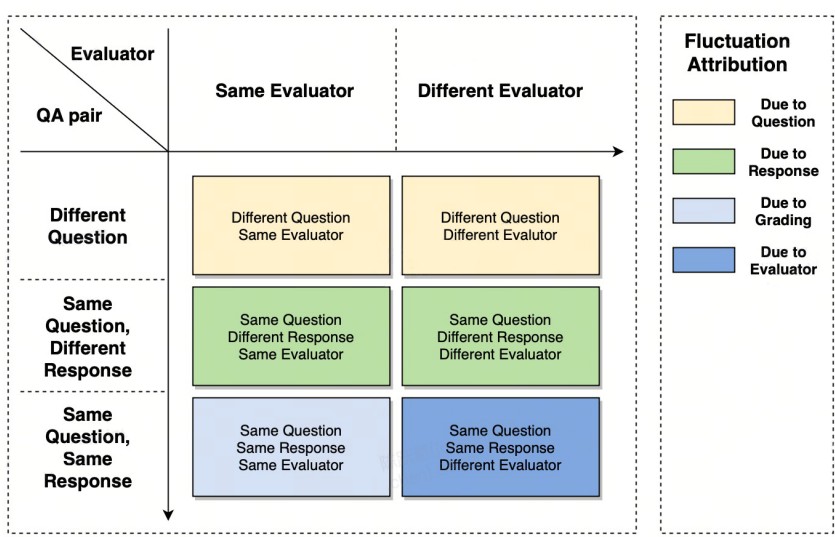

Figure 3: The six scenarios of single dimension grade breakdown

4. Calculating the contribution of the 4 fluctuation reasons to grade changes: Based on these 6 scenarios, the accuracy or grade of the LLM in a certain dimension can be broken down into 6 parts: *Accuracy of a certain dimension* = $W_{q1} * W_{p1} * Avg_{q1p1} + W_{q1} * W_{p2} * Avg_{q1p2} + W_{q2} * W_{p1} * Avg_{q2p1} + W_{q2} * W_{p2} * Avg_{q2p2} + W_{q3} * W_{p1} * Avg_{q3p1} + W_{q3} * W_{p2} * Avg_{q3p2}$, where q1, q2, q3 respectively

represent three scenarios: different questions, same questions and different responses, and same questions and same answers. p1, p2 respectively represent scenarios where the evaluators are the same or where the evaluators are different. Wq1 indicates the proportion of QA pairs where the questions are different from another round of evaluation. Wq2 indicates the proportion of QA pairs where the questions are the same as in another round of evaluation, but the responses are different. Wq3 indicates the proportion of QA pairs where both the question and the responses are the same as in another round of evaluation. Wp1 indicates the proportion of evaluators who are the same as those in another round of evaluation. Wp2 indicates the proportion of evaluators who are different from those in another round of evaluation. Avgq1p1 represents the average grade given by the same evaluator for QA pairs with different questions. Avgq1p2 represents the average grade given by different evaluators for QA pairs with different questions.

*Change in accuracy of a certain dimension* $= [W'_{q1} * W'_{p1} * Avg'_{q1p1} + W'_{q1} * W'_{p2} * Avg'_{q1p2} + W'_{q2} * W'_{p1} * Avg'_{q2p1} + W'_{q2} * W'_{p2} * Avg'_{q2p2} + W'_{q3} * W'_{p1} * Avg'_{q3p1} + W'_{q3} * W'_{p2} * Avg'_{q3p2}] - [W_{q1} * W_{p1} * Avg_{q1p1} + W_{q1} * W_{p2} * Avg_{q1p2} + W_{q2} * W_{p1} * Avg_{q2p1} + W_{q2} * W_{p2} * Avg_{q2p2} + W_{q3} * W_{p1} * Avg_{q3p1} + W_{q3} * W_{p2} * Avg_{q3p2}]$

$= \sum_{i=1}^{3} \sum_{j=1}^{2} (W'_{qi} W'_{pj} Avg'_{qipj} - W_{qi} W_{pj} Avg_{qipj})$

$= \sum_{i=1}^{3} \sum_{j=1}^{2} [(W'_{qi} W'_{pj} Avg'_{qipj} - W_{qi} W'_{pj} Avg'_{qipj}) + (W_{qi} W'_{pj} Avg'_{qipj} - W_{qi} W_{pj} Avg'_{qipj}) + (W_{qi} W_{pj} Avg'_{qipj} - W_{qi} W_{pj} Avg_{qipj})]$

$= \{[W_{q1} \sum_{j=1}^{2} W_{p1} (Avg'_{q1pj} - Avg_{q1pj})] + \sum_{i=1}^{3} \sum_{j=1}^{2} (W'_{qi} W'_{pj} Avg'_{qipj} - W_{qi} W'_{pj} Avg'_{qipj})\} \ldots \ldots (1)$

$+ \{W_{q2} \sum_{j=1}^{2} W_{p1} (Avg'_{q2pj} - Avg_{q2pj})\} \ldots \ldots (2)$

$+ \{W_{q3} W_{p1} (Avg'_{q3p1} - Avg_{q3p1})\} \ldots \ldots (3)$

$+ \{W_{q3} W_{p2} (Avg'_{q3p2} - Avg_{q3p2}) + \sum_{i=1}^{3} \sum_{j=1}^{2} (W_{qi} W'_{pj} Avg'_{qipj} - W_{qi} W_{pj} Avg'_{qipj})\} \ldots \ldots (4)$

If the three types of proportions of QA pairs (Wq) and the two types of proportions of evaluators (Wp) remain unchanged between two rounds of evaluations, then the above expression can be simplified to: *Change in accuracy of a certain dimension =*

$\{W_{q1} \sum_{j=1}^{2} W_{p1} (Avg'_{q1pj} - Avg_{q1pj})\} \ldots \ldots (1)$

$+ \{W_{q2} \sum_{j=1}^{2} W_{p1} (Avg'_{q2pj} - Avg_{q2pj})\} \ldots \ldots (2)$

$+ \{W_{q3} W_{p1} (Avg'_{q3p1} - Avg_{q3p1})\} \ldots \ldots (3)$

$+ \{W_{q3} W_{p2} (Avg'_{q3p2} - Avg_{q3p2})\} \ldots \ldots (4)$ As shown above, changes in grades can be attributed to: changes in questions (1), changes in responses (2), changes in grades by the same evaluators (3) and changes in evaluators (4).

# G  Details of Human Evaluator Training

1. *Selection of Human Evaluators:* Human evaluators should be selected from a candidate pool with domain expertise. The circumstances may vary among different domains. In our case, they were selected from our existing team of professional annotators who regularly perform domain-specific annotation tasks.

2. *Training Methodology:* Training sessions should be structured around the defined rubrics. Example-based training sessions should be conducted where annotators are exposed to diverse examples of responses and instructed on how to apply the evaluation criteria consistently.

3. *Trial Evaluations:* Initial trial evaluations should be conducted to assess the consistency within and among evaluators in their grading decisions. Issues identified during trials should be determined whether to be attributed to few evaluators based on measures described in Appendix F. If ambiguity found in grading rubrics, the expression of rubrics should be refined.

4. *Evaluation Quality Assurance:* For evaluators showing inconsistent performance, targeted retraining sessions were implemented. These sessions focused on reinforcing the rubrics and providing additional examples to clarify evaluation criteria. Post-retraining, evaluators should go through further trial evaluations to validate their improvement.

5. *Deployment Criteria:* Evaluators were only deployed into the production setting after achieving a predefined threshold of consistency in trial evaluations.

## H  Evaluation Results of All Capability Dimensions

| Capability | Capability Dimension | GPT-4 | Ernie Bot | PLLM3 | PLLM2 | PLLM1 |
|---|---|---|---|---|---|---|
| Domain | Conceptual and Terminological Understanding | 66.0 | 80.0 | 82.5 | 84.5 | 85.0 |
| Domain | Company Information | 22.5 | 85.0 | 92.0 | 79.0 | 98.5 |
| Domain | Legal and Policy Knowledge | 38.5 | 67.5 | 91.5 | 81.0 | 91.5 |
| Domain | Industry Insights | 48.0 | 84.5 | 88.5 | 85.5 | 90.0 |
| Domain | Company-specific Knowledge | 19.0 | 81.5 | 89.0 | 77.0 | 86.5 |
| Domain | Creative Capability in Logistics Context | 94.7 | 92.0 | 40.0 | 40.0 | 40.0 |
| General | Semantic Understanding | 98.0 | 92.0 | 90.0 | 82.0 | 88.0 |
| General | Contextual Conversation | 68.0 | 75.0 | 20.0 | 24.0 | 32.0 |
| General | Answer Completeness and Coherence | 64.0 | 90.0 | 54.0 | 56.0 | 54.0 |
| General | Factuality | 86.0 | 98.0 | 77.0 | 76.0 | 74.0 |
| General | Creativity | 70.0 | 81.3 | 55.3 | 60.7 | 65.3 |
| General | Logical Reasoning | 76.0 | 86.0 | 60.0 | 56.0 | 68.0 |

Table 9: The normalized average grades of evaluated LLMs by all capability dimensions

| Capability | Capability Dimension | Number of Annotators | Ratio of Disagreement |
|---|---|---|---|
| Domain | Conceptual and Terminological Understanding | 5 | 27.0% |
| Domain | Company Information | 5 | 17.0% |
| Domain | Legal and Policy Knowledge | 5 | 33.0% |
| Domain | Industry Insights | 5 | 24.0% |
| Domain | Company-specific Knowledge | 5 | 17.0% |
| Domain | Creative Capability in Logistics Context | 5 | 80.0% |
| General | Semantic Understanding | 5 | 22.0% |
| General | Contextual Conversation | 5 | 22.0% |
| General | Answer Completeness and Coherence | 5 | 10.0% |
| General | Factuality | 5 | 18.0% |
| General | Creativity | 5 | 40.0% |
| General | Logical Reasoning | 5 | 4.0% |

Table 10: The ratio of disagreement of evaluated LLMs by all capability dimensions

