# OpenReview forum: "LalaEval: A Holistic Human Evaluation Framework for Domain-Specific Large Language Models"
_colmweb.org/COLM/2024/Conference — COLM_

### Official Review · Reviewer_qpET · 2024-05-11

**Rating:** 4
**Confidence:** 4
**Ethics Flag:** 1

**Summary:**

1. This paper proposes LalaEval, a framework designed to standardized and systematic human evaluation for domain-specific LLMs. The framework consists of 5 components including domain specification, criteria setting, benchmark dataset creation, construction of evaluation rubrics, and analysis of evaluation outcomes.

2. The framework’s efficacy is demonstrated through a case study in the logistics industry, showing how it helps elucidate performance differences among LLMs.

**Reasons To Accept:**

1. Constructing a standardized framework for human evaluation processes is an intriguing idea. Relevant research can help formalize the assessment procedures in the field of large language models, leading to more reliable evaluation outcomes.

2. This study provides a detailed description of each module of LalaEval, comprehensively considering  the processes involved in the human evaluation procedure.

**Reasons To Reject:**

1. My major concern is that the experiments in this article struggle to demonstrate the advantages and effectiveness of the proposed evaluation framework. On one hand, the author conducted only a case study involving three models, lacking validation across a broader range of domains and models. Additionally, the experimental results presented do not clarify how LalaEval surpasses previous human evaluation paradigms. On the other hand, the author only displayed the final evaluation results and conclusions, lacking presentation and validation of intermediate steps. There is also a lack of detailed description of the human evaluation details, such as the number of evaluators involved, inter-annotator agreement, and so forth.

2. As an academic paper, I think that the framework proposed by this study does not present any fundamental innovation compared to the commonly used human evaluation processes previously established.

---

> ### Author Rebuttal · Authors · 2024-05-31
>
> Thank you for your valuable feedback. We have provided brief responses and will include any additional necessary details in camera-ready version.
> # Validation of More Models
> We present the overall grades for 6 additional models, 4 foundation and 2 fine-tuned.
> || GPT-3.5|ChatGLM-6B|Vicuna-13B|Qwen-7B|Fine-tuned Qwen-7B|Fine-tuned Baichuan-13B|
> |-|:-:|:-:|:-:|:-:|:-:|:-:|
> |Overall|46.70|33.18|20.22|65.40|71.62|57.11|
> # Validation of More Domains
> LalaEval is designed as a conceptual framework to evaluate domain-specific LLMs, implementing standardized end-to-end protocols of human evaluation, disentangled with any specific domain. We reported only logistics domain because we finalized LalaEval when exploring standardized human evaluation framework in this domain. Currently, we do successfully apply LalaEval now in other domains such as HR/IT maintanance/text-to-SQL/telemarketing and bring business value. We therefore believe those who are interested in employing LalaEval can easily implement it.
> # Intermediate Steps
> We present the results of 2 example out of 12 dimensions here due to character limit and provide complete results in further comments.
> | Capability Dimension|GPT4|Ernie Bot|PLLM3|PLLM2|PLLM1|
> |-|:-:|:-:|:-:|:-:|:-:|
> | Domain: Conceptual and Terminological Understanding|66.0|80.0|82.5|84.5|85.0|
> | General: Semantic Understanding|98.0|92.0|90.0|82.0|88.0|
> # Human Evaluation Details
> We had 5 annotators and the overall disagreement ratio is 23.8%. More details will be provided in further comments.
> # Contribution and Comparison to Previous Paradigms
> LalaEval is the first holistic human evaluation conceptual framework for domain-specific LLMs, addressing the lack of standardized protocols in existing literature.
> Given the open-generation nature of LLM and the high stakes of large-scale real-world domain-specific LLM deployments, LalaEval aims to reduce overall human-in-the-loop evaluation subjectivity, to ensure objective and aligned evaluation results and/or further guide the LLM development. Existing literature primarily mentions human evaluation of specific tasks without detailing implementation, standardization or even protocols (See Section 5.2 of arXiv:2307.03109). LalaEval fills this gap by proposing protocols for all five components and a systematic integration.
> Furthermore, our deployed framework in a real-world production setting demonstrates its practical effectiveness, contributing valuable LLM-related industry (esp. non-tech) insights.

---

> > ### Author Response · Authors · 2024-05-31
> >
> > # Intermediate Steps (Continued)
> > We take the opportunity here to further present the complete intermediate per-model evaluation results of all our 12 capability dimensions.
> > | Capability         | Capability Dimension                        | GPT4 | Ernie Bot | PLLM3 | PLLM2 | PLLM1 |
> > |--------------------|---------------------------------------------|:----:|:---------:|:-----:|:-----:|:-----:|
> > | Domain Capability  | Conceptual and Terminological Understanding | 66.0 |    80.0   |  82.5 |  84.5 |  85.0 |
> > | Domain Capability  | Company Information                         | 22.5 |    85.0   |  92.0 |  79.0 |  98.5 |
> > | Domain Capability  | Legal and Policy Knowledge                  | 38.5 |    67.5   |  91.5 |  81.0 |  91.5 |
> > | Domain Capability  | Industry Insights                           | 48.0 |    84.5   |  88.5 |  85.5 |  90.0 |
> > | Domain Capability  | Company-specific Knowledge                  | 19.0 |    81.5   |  89.0 |  77.0 |  86.5 |
> > | Domain Capability  | Creative Capability in Logistics Context    | 94.7 |    92.0   |  40.0 |  40.0 |  40.0 |
> > | General Capability | Semantic Understanding                      | 98.0 |    92.0   |  90.0 |  82.0 |  88.0 |
> > | General Capability | Contextual Conversation                     | 68.0 |    75.0   |  20.0 |  24.0 |  32.0 |
> > | General Capability | Answer Completeness and Coherence           | 64.0 |    90.0   |  54.0 |  56.0 |  54.0 |
> > | General Capability | Factuality                                  | 86.0 |    98.0   |  77.0 |  76.0 |  74.0 |
> > | General Capability | Creativity                                  | 70.0 |    81.3   |  55.3 |  60.7 |  65.3 |
> > | General Capability | Logical Reasoning                           | 76.0 |    86.0   |  60.0 |  56.0 |  68.0 |
> > # Human Evaluation Details (Continued)
> > We present here the number of annotators and our simplified disagreement analysis of the original five models. The result is at question-model-response level. We define a response as "disagreed" when not all five annotators consistently scored zero(bad response)/positive marks(good response) for each question-model-response.
> > | Capability         | Capability Dimension                        | Number of Annotators | Ratio of Disagreement |
> > |--------------------|---------------------------------------------|:--------------------:|:---------------------:|
> > | Domain Capability  | Conceptual and Terminological Understanding |           5          |         27.0%         |
> > | Domain Capability  | Company Information                         |           5          |         17.0%         |
> > | Domain Capability  | Legal and Policy Knowledge                  |           5          |         33.0%         |
> > | Domain Capability  | Industry Insights                           |           5          |         24.0%         |
> > | Domain Capability  | Company-specific Knowledge                  |           5          |         17.0%         |
> > | Domain Capability  | Creative Capability in Logistics Context    |           5          |         80.0%         |
> > | General Capability | Semantic Understanding                      |           5          |         22.0%         |
> > | General Capability | Contextual Conversation                     |           5          |         22.0%         |
> > | General Capability | Answer Completeness and Coherence           |           5          |         10.0%         |
> > | General Capability | Factuality                                  |           5          |         18.0%         |
> > | General Capability | Creativity                                  |           5          |         40.0%         |
> > | General Capability | Logical Reasoning                           |           5          |          4.0%         |

---

> > > ### Comment · Reviewer_qpET · 2024-06-05
> > >
> > > Thanks for response. However, my concerns about the novelty and its empirical comparison to other evaluation framework still exist. So I tend to keep my score.

---

### Official Review · Reviewer_Mk4S · 2024-05-11

**Rating:** 6
**Confidence:** 3
**Ethics Flag:** 1

**Summary:**

This paper introduces LalaEval, a holistic framework for conducting standardized human evaluations of domain-specific large language models (LLMs). The framework consists of five main components: domain specification, criteria establishment, benchmark dataset creation, construction of evaluation rubrics, and analysis and interpretation of evaluation outcomes. The paper demonstrates the framework's application within the logistics industry, presenting domain-specific evaluation benchmarks, datasets, and a comparative analysis of LLMs. The results highlight the framework's effectiveness in elucidating performance differences and guiding model selection and development for domain-specific LLMs.

**Reasons To Accept:**

- The paper addresses a crucial research gap by providing a systematic methodology for conducting standardized human evaluations within specific domains, which is essential for the practical application of LLMs in real-world business settings.
- The five-component framework is comprehensive and well-structured, covering all essential aspects of domain-specific LLM evaluation, from domain specification to the analysis and interpretation of evaluation results.
- The case study of the logistics industry demonstrates the framework's practical utility and its ability to generate valuable insights for businesses looking to integrate LLMs into their operations.

**Reasons To Reject:**

- The case study focuses on the logistics industry, which may limit the generalizability of the findings to other domains. Additional case studies across various industries would strengthen the paper's claims about the framework's broad applicability.
- The paper does not provide a detailed comparison of the LalaEval framework to existing evaluation methodologies, making it difficult to assess its relative advantages or disadvantages.

---

> ### Author Rebuttal · Authors · 2024-05-31
>
> Thank you for your valuable feedback. We have provided brief point-by-point responses below and will include any additional necessary details in the camera-ready version.
> # Generalizability to Other Domains
> LalaEval is designed as a conceptual framework to evaluate domain-specific LLMs, implementing standardized end-to-end protocols of human evaluation which are disentangled with any specific domain.  We reported only logistics domain in submission because we developed and finalized LalaEval when exploring standardized human evaluation framework in this domain. Currently, we do successfully apply LalaEval now in other domains such as HR/IT maintanance/text-to-SQL/telemarketing and bring business value. We therefore believe those who are interested in employing LalaEval to evaluate other domain-specific LLMs by human can easily implement it.
> # Contribution and Comparison to Previous Human Evaluation Paradigms
> LalaEval is the first holistic human evaluation conceptual framework for domain-specific LLMs, addressing the lack of standardized protocols in existing literature.
> Given the open-generation nature of LLM output and the high stakes of large-scale real-world domain-specific LLM deployments, LalaEval aims to reduce overall human-in-the-loop evaluation subjectivity, to ensure objective and aligned evaluation results and/or further guide our LLM development. Existing literature primarily mentions human evaluation of specific tasks without detailing implementation, standardization or even protocols (See Section 5.2 of A Survey on Evaluation of Large Language Models, arXiv:2307.03109 for a detailed review). LalaEval fills this gap by proposing protocols for domain specification, criteria/dataset/rubrics construction, result analysis including reliability measurement and a systematic integration of these components.
> Furthermore, our deployed framework in a real-world production setting demonstrates its practical effectiveness, contributing valuable LLM-related industry (esp. non-tech) insights.

---

### Official Review · Reviewer_uk9M · 2024-05-12

**Rating:** 5
**Confidence:** 3
**Ethics Flag:** 1

**Summary:**

This paper introduce a holistic framework for the human evaluation of domain specific large language models. This framework have five main components: domain specification, criteria establishment, benchmark dataset creation, construction of evaluation rubrics, analysis and interpretation of evaluation results.

**Questions To Authors:**

1.	How could we evaluate if the proposed LalaEval is working better than other evaluation metrics or frameworks?

2.	The font in the figures is a little small and it needs to be set larger.

**Reasons To Accept:**

1.	The evaluation framework for domain specific human evaluation of LLMs is beneficial to the community.

**Reasons To Reject:**

1.	Certain method details are missed. For example, it is hard to tell how to construct a hierarchical domain specification for the logistics industry in Figure 1. The authors mentioned about the methods of the principle of mutual exclusivity, qualitative prioritization and so on, yet these methods are not well described and would be hard to understand if the readers do not have related background.

2.	It would be better if the authors could release the dataset and evaluation code they’ve created in this work.

3.	There could be more analysis to the evaluation results. For example, GPT4 underperform all the compared language models in all the results. How could we explain this phenomenon and is there any reasons that lead this result?

---

> ### Author Rebuttal · Authors · 2024-05-31
>
> # Contribution and Comparison to Previous Human Evaluation Paradigms
> LalaEval is the first holistic human evaluation conceptual framework for domain-specific LLMs, addressing the lack of standardized protocols in existing literature.
> Given the open-generation nature of LLM output and the high stakes of large-scale real-world domain-specific LLM deployments, LalaEval aims to reduce overall human-in-the-loop evaluation subjectivity, to ensure objective and aligned evaluation results and/or further guide our LLM development. Existing literature primarily mentions human evaluation of specific tasks without detailing implementation, standardization or even protocols (See Section 5.2 of A Survey on Evaluation of Large Language Models, arXiv:2307.03109 for a detailed review). LalaEval fills this gap by proposing protocols for domain specification, criteria/dataset/rubrics construction, result analysis including reliability measurement and a systematic integration of these components.
> Furthermore, our deployed framework in a real-world production setting demonstrates its practical effectiveness, contributing valuable LLM-related industry (esp. non-tech) insights.
> # Importance of Human Evaluation
> The domain-specific LLM deployment typically has high-stake real-world influence (e.g. chatbot serving millions of consumers) and the LLM response has open-generation nature, so it is highly crucial to involve human evaluation in the loop to ensure alignment and tailored to domain needs among other goals. Although automated methods may serve as an important supplement to the human evaluation, it is beyond the scope of this paper.

---

> > ### Comment · Reviewer_uk9M · 2024-06-02
> > **Response to the authors**
> >
> > I feel much sorry that I attached the wrong review to this submission, I have changed the reviews accordingly. Sorry again for my bad mistake.

---

> > > ### Author Response · Authors · 2024-06-03
> > > **Rebuttal of the New Review**
> > >
> > > Thank you for your new valuable feedback. We have provided brief point-by-point responses below and will include any additional necessary details in the camera-ready version.
> > > # Method Elaboration
> > > We will emphasize that domain specification is primarily conceptual and qualitative, relying on expert knowledge and understanding of the business context. Specifically, we will:
> > > 1. Explain the conceptual principles used:
> > >       - Mutual exclusivity: each subdomain should be distinct and non-overlapping
> > >       - Collective exhaustiveness: all relevant aspects of the domain should be covered
> > > 2. Elaborate on the qualitative prioritization process:
> > >       - Priorities are assigned based on expert judgment of subdomain relevance
> > >       - Consideration is given to alignment with specific business objectives and focus areas
> > >       - The prioritization is conducted through discussions with domain experts and stakeholders (e.g. managers of the relevant businesses)
> > > 3. Provide a step-by-step guide on applying this qualitative methodology in the example logistics industry context:
> > >       - Begin with identifying the most granular, relevant subdomains based on expert knowledge
> > >       - Iteratively combine these granular subdomains into broader, more comprehensive categories
> > >       - Assign priorities to each subdomain and category based on their determined relevance and business alignment
> > >       - Continue this process until a sufficiently comprehensive domain specification is constructed
> > > 4. Include concrete examples to illustrate each step of the conceptual process for a general audience:
> > >       - Demonstrate how "Intracity Freight Transportation" is identified as the most granular, highest priority subdomain
> > >       - Show how it is combined with other subdomains like "Intercity Freight Transportation" into the broader "Road Transportation" category
> > >       - Explain how priority levels are assigned qualitatively based on Huolala's business focus and expert judgment
> > > # Opensource Request
> > > Our request to open-source is currently going through the internal approval process. We will release the dataset and code once it is approved.
> > > # Lower Scores of GPT-4
> > > From the detailed results of all 12 capability dimensions, the lower scores of the foundation model GPT-4 are likely due to the absence of proprietary information used to fine-tune the PLLMs. They performed better on general capabilities.
> > > | Capability         | Capability Dimension                        | GPT4 | Ernie Bot | PLLM3 | PLLM2 | PLLM1 |
> > > |--------------------|---------------------------------------------|:----:|:---------:|:-----:|:-----:|:-----:|
> > > | Domain Capability  | Conceptual and Terminological Understanding | 66.0 |    80.0   |  82.5 |  84.5 |  85.0 |
> > > | Domain Capability  | Company Information                         | 22.5 |    85.0   |  92.0 |  79.0 |  98.5 |
> > > | Domain Capability  | Legal and Policy Knowledge                  | 38.5 |    67.5   |  91.5 |  81.0 |  91.5 |
> > > | Domain Capability  | Industry Insights                           | 48.0 |    84.5   |  88.5 |  85.5 |  90.0 |
> > > | Domain Capability  | Company-specific Knowledge                  | 19.0 |    81.5   |  89.0 |  77.0 |  86.5 |
> > > | Domain Capability  | Creative Capability in Logistics Context    | 94.7 |    92.0   |  40.0 |  40.0 |  40.0 |
> > > | General Capability | Semantic Understanding                      | 98.0 |    92.0   |  90.0 |  82.0 |  88.0 |
> > > | General Capability | Contextual Conversation                     | 68.0 |    75.0   |  20.0 |  24.0 |  32.0 |
> > > | General Capability | Answer Completeness and Coherence           | 64.0 |    90.0   |  54.0 |  56.0 |  54.0 |
> > > | General Capability | Factuality                                  | 86.0 |    98.0   |  77.0 |  76.0 |  74.0 |
> > > | General Capability | Creativity                                  | 70.0 |    81.3   |  55.3 |  60.7 |  65.3 |
> > > | General Capability | Logical Reasoning                           | 76.0 |    86.0   |  60.0 |  56.0 |  68.0 |

---

> > > > ### Author Response · Authors · 2024-06-03
> > > > **Rebuttal of the New Review (Continued)**
> > > >
> > > > # Contribution and Comparison to Previous Framework
> > > > There is no standardized end-to-end (from domain specification to criteria/dataset/rubrics construction to result interpretation) human evaluation protocols that meet industry-level domain needs currently, scoring metrics and and corresponding interpretation are only part of LalaEval, so LalaEval is the first holistic human evaluation conceptual framework for domain-specific LLMs, addressing the lack of standardized protocols in existing literature and it is hard to be compared to previous paradigms. Given the open-generation nature of LLM output and the high stakes of large-scale real-world domain-specific LLM deployments, LalaEval aims to reduce overall human-in-the-loop evaluation subjectivity, to ensure objective and aligned evaluation results and/or further guide our LLM development. Existing literature primarily mentions human evaluation of specific tasks without detailing implementation, standardization or even protocols (See Section 5.2 of A Survey on Evaluation of Large Language Models, arXiv:2307.03109 for a detailed review). LalaEval fills this gap by proposing protocols for domain specification, criteria/dataset/rubrics construction, result analysis including reliability measurement and a systematic integration of these components. Furthermore, our deployed framework in a real-world production setting demonstrates its practical effectiveness, contributing valuable LLM-related industry (esp. non-tech) insights.
> > > > # Font Size
> > > > Thank you for pointing out. We will fix the font size once we are allowed to submit a revision.

---

### Official Review · Reviewer_dWK1 · 2024-05-20

**Rating:** 5
**Confidence:** 4
**Ethics Flag:** 1

**Summary:**

The paper introduces LALAEval, a comprehensive human evaluation framework designed to assess LLMs. The framework incorporates a diverse set of evaluation tasks and metrics to provide a holistic understanding of LLM's performance, addressing the limitations of existing evaluation methods.

**Questions To Authors:**

1. I was questioned about the adapability of this framework to various domain applications.
2. what is the novelty of this proposed framework?

**Reasons To Accept:**

1. The framework encompasses a diverse range of evaluation tasks and metrics, allowing for a nuanced assessment of LLMs' performance.
2. The empirical results showcase the efficacy of LALAEval in capturing various aspects of LALMs' behavior and facilitating fair model comparisons.

**Reasons To Reject:**

The discussion on potential biases or limitations of the proposed framework, as well as strategies to mitigate them, could be more comprehensive.

---

> ### Author Rebuttal · Authors · 2024-05-31
>
> # Discussion on Potential Biases or Limitations
> We will include more details in the camera-ready version. Briefly, LalaEval faces potential limitations such as evaluator subjectivity, data selection bias, dynamic domain knowledge changes, and scalability issues. To mitigate these, we propose implementing standardized training programs with regular assessments for evaluators. Ensuring a representative benchmark dataset can reduce selection bias, while continuous updates will keep the framework current with industry changes. Additionally, supplementing automated methods and support systems can streamline the evaluation process and enhance scalability and reliability.
> # Adaptability of LalaEval to More Domains
> LalaEval is designed as a conceptual framework to evaluate domain-specific LLMs, implementing standardized end-to-end protocols of human evaluation which are disentangled with any specific domain.  We reported only logistics domain in submission because we developed and finalized LalaEval when exploring standardized human evaluation framework in this domain. Currently, we do successfully apply LalaEval now in other domains such as HR/IT maintanance/text-to-SQL/telemarketing and bring business value. We therefore believe those who are interested in employing LalaEval to evaluate other domain-specific LLMs by human can easily implement it.
> # Novelty of LalaEval
> LalaEval is the first holistic human evaluation conceptual framework for domain-specific LLMs, addressing the lack of standardized protocols in existing literature.
> Given the open-generation nature of LLM output and the high stakes of large-scale real-world domain-specific LLM deployments, LalaEval aims to reduce overall human-in-the-loop evaluation subjectivity, to ensure objective and aligned evaluation results and/or further guide our LLM development. Existing literature primarily mentions human evaluation of specific tasks without detailing implementation, standardization or even protocols (See Section 5.2 of A Survey on Evaluation of Large Language Models, arXiv:2307.03109 for a detailed review). LalaEval fills this gap by proposing protocols for domain specification, criteria/dataset/rubrics construction, result analysis including reliability measurement and a systematic integration of these components.
> Furthermore, our deployed framework in a real-world production setting demonstrates its practical effectiveness, contributing valuable LLM-related industry (esp. non-tech) insights.

---

### Official Review · Reviewer_wsJT · 2024-05-23

**Rating:** 6
**Confidence:** 3
**Ethics Flag:** 1

**Summary:**

This paper proposes LalaEval, an human-centered evaluation framework for domain-specific large language models (LLMs). The core principles behind the proposed work consist of five components: domain specification, criteria establishment, benchmark creation, evaluation rubric construction and results interpretation. This work combines the proposed principles and presents a general deployable framework. The authors also present the evaluation results of 3 proprietary models fine tuned on top of ChatGLM along with Ernie Bot, GPT-4. Overall this paper is more of an industry-track position piece than a pure methodological one.

**Questions To Authors:**

For the main evaluation, should I believe that the lower scores of GPT-4 and Ernie Bot are due to the fact they do not have access to potentially proprietary information that is used to fine-tune the in-house ChatGLM? Additionally, how does LalaEval handle the potential subjectivity and variability in human evaluations, and what measures are in place to ensure the consistency and reliability of these evaluations?

**Reasons To Accept:**

This work is well motivated and I agree with the authors that LLMs, especially in domain-sensitive scenarios, need to have better testing protocols. This paper presents a conceptual framework that allows systematic integration of human judgment in evaluating LLM’s in-domain capabilities. From my perspective, the overall framework seems to be well structured and can be standardized in real-world applications.

**Reasons To Reject:**

From my understanding, LalaEval seems to demand significant resources, including time, trained personnel, and high-quality domain-specific data. Despite the nature of this piece being a positional one, I feel that this concept will be more complete if the authors can provide more details on the evaluations. Specifically, slightly more information on the process of selecting human evaluators, the training they receive, and how their evaluations are standardized would be beneficial.

Additionally, as the authors have outlined in the evaluation framework constructions, there are multiple scoring stages. It would be beneficial to include them in the main evaluation for a more detailed per-model evaluation.

Minor suggestions:
- Try increasing the font size of figure 1 or otherwise the figure can be difficult to read.
- Be careful about the use of \citep and \citet (e.g. first sentence of Related Work)
- When describing framework specifications, avoid using specific examples as the bullet points (e.g., change "Specific Knowledge about
  Huolala" to "Company-specific Knowledge" in 3.3.2).
- If possible, consider open-sourcing a prototype.

---

> ### Author Rebuttal · Authors · 2024-05-31
>
> Thank you for your valuable feedback. We have provided brief point-by-point responses below and will include any additional necessary details in the camera-ready version.
> # Resource Demand
> The domain-specific LLM deployment typically has large real-world influence (e.g. chatbot serving millions of consumers) and has significant impact on firm operation, so it is highly crucial to evaluate and benchmark domain-specific LLMs. In these widespread high-stake circumstances, the resources required are affordable compared to the potential benefits of ensuring best practice.
> # More Details on Evaluation
> 1. In our case, the human evaluators were selected from our existing professional annotator team who regularly handle domain-specific annotation tasks.
> 2. Our training was based on our framework's rubrics and example-based training sessions. Besides, we conducted trial evaluations to identify evaluator variability. If the variation was attributed to few evaluators, we would retrain them with rubrics and more examples and then reran trial evaluations to form a local loop. Only after reaching an acceptable threshold, they were deployed into the production setting.
> 3. We included solving variability in Appendix F in our submission. We standardize the disagreement and fluctuation and attribute it to model, data, intra-evaluators or inter-evaluators. These quantitative measures of reliability were integrated into the loop of dataset construction, model development, training and interpretation.
> # Detailed Evaluation Results
> We present the results of 2 example out of 12 dimensions here due to character limit and provide complete results in further comments.
> | Capability Dimension                        | GPT4 | Ernie Bot | PLLM3 | PLLM2 | PLLM1 |
> |-|:-:|:-:|:-:|:-:|:-:|
> | Domain: Conceptual and Terminological Understanding | 66.0 |    80.0   |  82.5 |  84.5 |  85.0 |
> | General: Semantic Understanding                      | 98.0 |    92.0   |  90.0 |  82.0 |  88.0 |
> # Minor Suggestion
> We will fix them. Our request to open-source is currently going through the internal approval process.
> # Lower Scores of GPT-4 and Ernie Bot
> From the detailed results, the lower scores of GPT-4 and Ernie Bot are likely due to the absence of proprietary information used to fine-tune the PLLMs.
> # Handling Subjectivity and Variability
> Please refer to rebuttal 2(3) and the details are included in Appendix F. We will also include the results of reliability measures in the camera-ready version.

---

> > ### Author Response · Authors · 2024-05-31
> >
> > # Detailed Evaluation Results (Continued)
> > We take the opportunity here to further present the complete intermediate per-model evaluation results of all our 12 capability dimensions.
> > | Capability         | Capability Dimension                        | GPT4 | Ernie Bot | PLLM3 | PLLM2 | PLLM1 |
> > |--------------------|---------------------------------------------|:----:|:---------:|:-----:|:-----:|:-----:|
> > | Domain Capability  | Conceptual and Terminological Understanding | 66.0 |    80.0   |  82.5 |  84.5 |  85.0 |
> > | Domain Capability  | Company Information                         | 22.5 |    85.0   |  92.0 |  79.0 |  98.5 |
> > | Domain Capability  | Legal and Policy Knowledge                  | 38.5 |    67.5   |  91.5 |  81.0 |  91.5 |
> > | Domain Capability  | Industry Insights                           | 48.0 |    84.5   |  88.5 |  85.5 |  90.0 |
> > | Domain Capability  | Company-specific Knowledge                  | 19.0 |    81.5   |  89.0 |  77.0 |  86.5 |
> > | Domain Capability  | Creative Capability in Logistics Context    | 94.7 |    92.0   |  40.0 |  40.0 |  40.0 |
> > | General Capability | Semantic Understanding                      | 98.0 |    92.0   |  90.0 |  82.0 |  88.0 |
> > | General Capability | Contextual Conversation                     | 68.0 |    75.0   |  20.0 |  24.0 |  32.0 |
> > | General Capability | Answer Completeness and Coherence           | 64.0 |    90.0   |  54.0 |  56.0 |  54.0 |
> > | General Capability | Factuality                                  | 86.0 |    98.0   |  77.0 |  76.0 |  74.0 |
> > | General Capability | Creativity                                  | 70.0 |    81.3   |  55.3 |  60.7 |  65.3 |
> > | General Capability | Logical Reasoning                           | 76.0 |    86.0   |  60.0 |  56.0 |  68.0 |

---

### Author Response · Authors · 2024-06-07

As the discussion period is about to end, we would like to extend our sincere thanks to all the reviewers for their valuable feedback. We also want to take this opportunity to reiterate our main contributions:

1. **Standardizing the Evaluation Process**: Compared to the current literature that often involves collecting datasets without a structured approach and inviting domain experts to score subjectively, we proposed a meta-framework to standardize the end-to-end process for evaluating domain-specific LLMs. Our framework addresses key questions such as, “How shall we define our domain?”, “How shall we set capability criteria and build our own evaluation datasets/benchmarks?”, and “How shall we train domain annotators and analyze their evaluations to minimize subjectivity?”.

2. **Successful Deployment and Impact**: We have successfully deployed our framework within the model selection and development process in our domain. The effectiveness of our framework is proven to be crucial to the implementation of domain-specific LLMs that serve millions of consumers. Furthermore, the deployment of our framework in other domains such as HR/IT maintenance/text-to-SQL/telemarketing has also made significant progress , demonstrating its versatility and impact. We believe that industry insights gained from large-scale real-world evaluation deployment can bring unique value to the community.

Thank you once again for your constructive feedback and consideration.

---

### Decision · Program_Chairs · 2024-07-10

**Decision:**

Accept

**Comment:**

The paper outlines a general framework for designing evaluations for domain-specific LLMs. It appears to be an industry paper, which is grounded in practical concerns about deploying LLMs for real-world use cases. As such, it does not follow the “typical” style of an academic paper, and this is a cause for some reviewer concerns. In particular, reviewers raise concerns about a lack of clear evaluation/baselines for existing approaches, and about the apparent cost of carrying out the proposed framework. The authors’ response is that no clear baselines exist because industry standards do not yet exist, and rather each case is handled in an ad hoc manner. This ad hoc nature of evaluation is the problem the paper attempts to address.

I believe that this paper would be a useful addition to COLM precisely because of its atypical nature. The reviewers’ concerns about evaluation are understandable, but given the current state of LLM research, there is likely more to be gained by exposing differing approaches and perspectives (e.g., from academia and industry, from quantitative and qualitative sciences) than prematurely imposing a specific recipe for research. This paper has value in outlining the key challenges that are faced by those applying LLMs in industry settings, and showcasing the way that these challenges are currently addressed. Such work could provide inspiration or fodder for subsequent, more rigorous, research studies.

[At least one review was discounted during the decision process due to quality]